# Characterization of GABA-Transaminase Gene from Mulberry (*Morus multicaulis*) and Its Role in Salt Stress Tolerance

**DOI:** 10.3390/genes13030501

**Published:** 2022-03-12

**Authors:** Mengru Zhang, Zhaoyang Liu, Yiting Fan, Chaorui Liu, Hairui Wang, Yan Li, Youchao Xin, Yingping Gai, Xianling Ji

**Affiliations:** 1State Key Laboratory of Crop Biology, Shandong Agricultural University, Taian 271018, China; 15646708303@163.com (M.Z.); yqwsdau@163.com (Y.F.); 2College of Forestry, Shandong Agricultural University, Taian 271018, China; lzysdau@163.com (Z.L.); linxueyuan217@163.com (C.L.); gypsd@sina.com (H.W.); sdau738dq@163.com (Y.L.); ycxin@sdau.edu.cn (Y.X.)

**Keywords:** mulberry, γ-aminobutyric acid, GABA-transaminase, salt stress, reactive oxygen species

## Abstract

Gamma-aminobutyric acid (GABA) has been reported to accumulate in plants when subjected to salt stress, and GABA-transaminase (GABA-T) is the main GABA-degrading enzyme in the GABA shunt pathway. So far, the salt tolerance mechanism of the *GABA-T* gene behind the GABA metabolism remains unclear. In this study, the cDNA (designated *Mu**GABA-T*) of *GABA-T* gene was cloned from mulberry, and our data showed that MuGABA-T protein shares some conserved characteristics with its homologs from several plant species. *MuGABA-T* gene was constitutively expressed at different levels in mulberry tissues, and was induced substantially by NaCl, ABA and SA. In addition, our results demonstrated that exogenous application of GABA significantly reduced the salt damage index and increased plant resistance to NaCl stress. We further performed a functional analysis of *MuGABA-T* gene and demonstrated that the content of GABA was reduced in the transgenic *MuGABA-T Arabidopsis* plants, which accumulated more ROS and exhibited more sensitivity to salt stress than wild-type plants. However, exogenous application of GABA significantly increased the activities of antioxidant enzymes and alleviated the active oxygen-related injury of the transgenic plants under NaCl stress. Moreover, the *MuGABA-T* gene was overexpressed in the mulberry hairy roots, and similar results were obtained for sensitivity to salt stress in the transgenic mulberry plants. Our results suggest that the *MuGABA-T* gene plays a pivotal role in GABA catabolism and is responsible for a decrease in salt tolerance, and it may be involved in the ROS pathway in the response to salt stress. Taken together, the information provided here is helpful for further analysis of the function of *GABA-T* genes, and may promote mulberry resistance breeding in the future.

## 1. Introduction

Plants are exposed to a wide array of stresses in their natural surroundings, among which salt stress is a major abiotic stress factor, which has negative effects on plant growth and production. It was estimated that more than 20% of agricultural lands worldwide are afflicted by salinity, and soil salinization has become a major constraint for agricultural production [1,2]. In the long evolutionary progress, plants have developed sophisticated mechanisms that attenuate the adverse effects of salt stress. It has been well documented that plants may accumulate compatible solutes, such as carbohydrates and amino acids, which are energy source, osmotic regulants, as well as signaling molecules under saline conditions [3,4]. Among these stress-responsive metabolites, γ-aminobutyric acid (GABA) is of special interest since its anabolic metabolism could be activated by salt stress, and it has been observed to be accumulated rapidly in the plants under salt stress [5,6,7,8]. Previous studies have demonstrated that GABA has an important effect on plant salt tolerance as a signal substance or temporary nitrogen pool, or as a regulator to cytoplasmic pH and antioxidant responses [8,9].

It was shown that the functions of GABA in plants are involved in the short metabolic pathway described as the GABA shunt pathway [9,10,11]. In this process, α-ketoglutarate takes glutamate as a substrate, forms GABA in mitochondria under the catalysis of glutamate decarboxylase (GAD), then converts into succinic semialdehyde (SSA) under the catalysis of GABA transaminase (GABA-T) and finally is converted to succinic acid by SSA dehydrogenase (SSADH), which is added to the TCA cycle or is transported back to the cytoplasm [12,13,14]. Among the GABA shunt enzymes, GAD is the most sensitive one in response to abiotic stress, and its gene expression and enzyme activity levels are closely associated with the enhancement of plant stress resistance mediated by GABA [6,15,16]. In wheat (*Triticum aestivum*), tobacco (*Nicotiana sylvestris*), melon (*Cucumis melo*) and other plants, the expression levels of *GAD* genes were significantly enhanced under salt stress, resulting in the increase in GABA content and high salt tolerance [7,17,18]. However, the content of GABA in plant tissues could not be controlled only by its synthesis rate, but also by its degradation rate [14,19,20]. Recent studies have shown that GABA-T is also a crucial factor in GABA shunt pathway. In the *GABA-T*-deficient mutants (*gaba-t* or *pop2*), GABA catabolism was blocked and high levels of GABA were found in the tissues [21]. It was reported that the *pop2−1* mutant is resistant to E-2-hexenal and the accumulation of alanine in roots is very low under hypoxia [21]. Consistent with this, it was found that the *pop2−1* mutant contained higher endogenous GABA and showed reduced aluminum toxicity. Similarly, in the triple mutant *gad1/gad2 × pop2-5*, the content of endogenous GABA was also increased, and it was observed to be less sensitive to drought stress [9,22,23,24]. However, Renault (2010) reported that although more GABA accumulated in the roots of pop2 than in those of the wild-type plants, the metabolic damage of GABA led to their oversensitivity to ion stress and reduced salt tolerance [25]. However, contrary to this result, Su et al. (2019) found that the *pop2-5* mutants, which were able to accumulate more GABA in their roots, were more resistant to salt stress than wild-type plants [26]. However, only a few studies have examined the relationships between *GABA-T* mutants, GABA and salt tolerance, so far, and the salt tolerance mechanism of the *GABA-T* gene behind the GABA metabolism has not been elucidated clearly.

Mulberry tree (*Morus* L.) is an economically important perennial woody plant, which has multiple uses not only in silkworm rearing, but has also been used extensively in ecology, pharmaceuticals and traditional Chinese medicines [27]. Mulberry can adapt well to adverse abiotic stresses, especially saline condition, but the molecular mechanisms responsible for salt stress have not been fully investigated [28]. Given that the *GABA-T* gene plays a pivotal role in the GABA metabolic process and mediating stress response, there has been no information concerning the *GABA-T* gene in mulberry to date. In the present study, the *GABA-T* gene of mulberry was cloned and molecularly characterized, and its expression profiles and functions in regulating defense responses against salt stresses were analyzed. These findings have laid the groundwork for studies to further explore the function and the mechanism of *GABA-T* gene in salt resistance.

## 2. Materials and Methods

### 2.1. Biological Materials

Mulberry (Husang 32 (*Morus multicaulis*) and Guisang You 62 (*M. atropurpurea*)), *N. benthamiana* and *A**. thaliana* (Col-0) plants were planted in the greenhouse at 26 and 22 °C, respectively, with circadian rhythms of 16 h light/8 h dark and humidity of 50–60%.

### 2.2. Cloning and Sequence Analysis

Samples used for RNA extraction were ground to a fine powder in liquid nitrogen using a mortar and pestle, and then RNA was extracted with TRIzol^®^ RNA Isolation Reagent (Invitrogen, Carlsbad, CA, USA) according to the manufacturer’s instructions, including an optional cleaning step using RNase-free DNase to eliminate genomic DNA contamination. The extracted RNA was quantified using a NanoDrop (Thermo, Waltham, MA, USA) and assessed for integrity on a 2100 Agilent Bioanalyzer (Agilent, 5301 Stevens Creek Boulevard, Santa Clara, CA, USA) and then was reverse transcribed with 100 units of M-MLV reverse transcriptase (Promega, Madison, WI, USA) in 20 mL reactions. The specific PCR primers (forward primer: 5′-TTCAAAGACCTCCTCTGTTC-3′; reverse primer: 5′-ATCACCATCATTCACTACG-3′) were designed based on our available mulberry transcriptome data for PCR amplification, and the PCR products were subsequently purified and sequenced. The predicted amino acid sequence of the gene was aligned with GABA-T family members from other plants, and a phylogenetic tree was constructed based on the protein sequences using the DNAMAN (version 5.2.2) program. Protein structure was predicted with online tools (http://swiss-model.expasy.org/, accessed on 19 January 2022).

### 2.3. Quantitative Real-Time-PCR Analysis

The primers (forward primer: 5′-GCACTTGCTGGTCTTTGGTG-3′; reverse primer: 5′-AGTGTCGTTGGCATCTGACC-3′) were designed for the quantitative real-time-PCR (qRT-PCR), which was performed with the SYBR Premix Ex Taq™ kit (Takara, Shanghai, China) on the CFX96TM Real-time System (Bio-Rad Laboratories, Hercules, CA, USA). The *ACTIN* gene (forward primer: 5′-CACTGAGGCTCCTTTGAACCC-3′; reverse primer: 5′-AGGTCGAGACGGAGAATAGCATG-3′) was used as an endogenous control to quantify the gene expression levels with the comparative Ct method [29]. All samples were analyzed in triplicate.

### 2.4. Promoter Activity Analysis

The promoter of the gene was cloned with specific primers (forward primer: 5′-CATTGTTCAAGGGACGAGA-3′; reverse primer: 5′-GAGAGAGGAGAGAGGAGTGTGA-3′) to replace the *35S* promoter in the binary vector pBI121 and was fused with the *GUS* gene to create the promoter expression vector. Then, the vector obtained was introduced into *A. tumefaciens* (strain GV3101) and used to infiltrate *N. benthamiana* leaves. The infiltrated leaves were treated with abscisic acid (ABA), salicylic acid (SA), methyl jasmonate (MeJA) and NaCl by spraying 100 μmol·L^−1^ ABA, 1 mol·L^−1^ SA or 25 μmol·L^−1^ MeJA or 50 mmol·L^−1^ NaCl solutions on the leaves. Expression of β-glucuronidase (GUS) gene in the infiltrated leaves was assessed by histochemical staining [30].

### 2.5. Generation of Transgenic Plants

Transgenic *Arabidopsis* plants were produced with the *Agrobacterium*-mediated floral dip transformation method [31]. Transgenic *Arabidopsis* seeds were stratified on MS plates supplemented with 50 μg·mL^−1^ kanamycin, and the T_3_ homozygous lines were selected for further experiments.

For production of hairy root transgenic mulberry plants, the gene was cloned into the binary vector pROK2 and was then introduced into *A. rhizogenes* K599 strains. The Guisang You 62 mulberry seedlings with the first pair of leaves were used for agro-infiltration as described previously by [32]. Four weeks later, the well-developed hairy roots were detected by PCR and the original roots were cut off, and the plants with positive hairy roots were used for subsequent experiments.

### 2.6. Plant Treatment

To determine the effect of GABA on mulberry salt tolerance, two-month-old mulberry plantlets under a regular watering regime were watered with 0‰ or 4‰ NaCl solution, respectively. At the same time, 50 mmol·L^−1^ GABA solution was sprayed on the leaves for 7 consecutive days, and the plantlets sprayed with water were used as controls. All the plantlets were kept in a greenhouse at 26 °C with a 16 h light/8 h dark photoperiod.

To determine the effect of GABA on *Arabidopsis* salt tolerance, two types of phenotyping experiments were conducted, for seeds sown on agar medium or seedlings grown in soil. In the former case, surface-sterilized seeds were sown on 1/2 MS medium supplemented with NaCl (0, 100 and 150 mmol·L^−1^ NaCl) or together with 5 mmol·L^−1^ GABA. After vernalization at 4 °C for 2 d, they were placed in the greenhouse as described above, and the germination rates were measured on the 7th day of germination. At the same time, some Petri dishes were oriented vertically for 1 week to measure the root length. For pot phenotyping experiments, four-week-old seedlings were watered with 0, 100 and 150 mmol·L^−1^ NaCl solution, respectively. At the same time, 50 mmol·L^−1^ GABA solution was sprayed on the leaves for 7 consecutive days, and the seedlings sprayed with water were used as controls. All the experiments above were tested in triplicate and repeated three times independently.

### 2.7. Hydrogen Peroxide, Superoxide and Enzyme Activity Assays

Superoxide (O_2_^−^) and hydrogen peroxide (H_2_O_2_) in the leaves were detected by nitroblue tetrazolium (NBT) and 3,3′-diaminobenzidine (DAB) staining, respectively, according to the methods described before [33]. For enzyme activity assays, the fresh leaves were powdered and then extracted followed the steps described before [33]. The superoxide dismutase (SOD), peroxidase (POD) and catalase (CAT) activities were measured according to the methods described by Beyer and Fridovich (1987) [34], Hemeda and Klein (1990) [35] and Aebi (1983) [36], respectively. All the experiments were performed in triplicate and repeated three times independently.

### 2.8. Measurements of GABA and Malondialdehyde

GABA content was measured with ELISA Kit. The Assay Kit (Art. No. G1106F) (Shanghai Enzyme-linked Biotechnology, Shanghai, China) was used following the manufacturer’s instructions. To measure the malondialdehyde (MDA) content, the samples were homogenized in trichloroacetic acid buffer (5% *w*/*v*), then centrifuged at 12,000× *g* at 4 °C for 15 min. Aliquot of the supernatant was collected and mixed with thiobarbituric acid (0.5% *w*/*v*) and boiled (100 °C) for 25 min. Then the mixture was centrifuged (7500× *g*) for 5 min to collect the supernatant, which was used for MDA concentration measurement following the methods described by Peever and Higgins (1989) [37]. All measurements were repeated three times.

### 2.9. Statistical Analysis

The text data obtained were analyzed using Microsoft Excel software, and the variance analysis and significance relationships were analyzed with SPSS 26.0 software (IBM, Armonk, NY, USA) at *p* ≤ 0.05.

## 3. Results

### 3.1. Exogenous Application of GABA Conferred Increased Tolerance of Salt Stress on Mulberry Seedlings

In order to explore whether exogenous GABA affects the salt tolerance of mulberry seedlings, the seedlings treated with 4‰ NaCl were sprayed with GABA for 7 consecutive days. Two weeks later, the seedlings sprayed with water (control) and 50 mmol·L^−1^ GABA showed little difference in the symptoms under the normal conditions. However, when they were exposed to NaCl stress, all of the seedlings displayed severe growth retardation, and a rapid wilting emergence of older leaves appeared in the control seedlings. Whereas the salt damage symptoms of the mulberry seedlings sprayed with GABA were significantly reduced, and only the old leaves turned yellow and the young leaves showed only slight wilting symptoms (Figure 1A). These results showed that the exogenous GABA alleviated the damage symptoms of mulberry under salt stress. To confirm this, the MDA contents in the leaves were analyzed, and the results showed that the MDA accumulations were increased in all the seedlings under salt stress, but the plants treated with GABA showed significantly lower MDA content than the control ones (Figure 1B). Moreover, the accumulation of H_2_O_2_ and O_2_^−^ in the seedlings was also examined. The data indicated that the accumulation of H_2_O_2_ showed no difference between the leaves sprayed with GABA or water under the normal conditions. Although the H_2_O_2_ level was markedly increased in leaves of the seedlings under salt stress, the accumulation of H_2_O_2_ in the leaves of controls was significantly higher than that sprayed with GABA (Figure 1C). The accumulation of O_2_^−^ in the leaves showed a similar pattern as the H_2_O_2_ accumulation (Figure 1D). Therefore, exogenous application of GABA might lead to enhanced reactive oxygen scavenging ability and confer increased tolerance of salt stress on the mulberry seedlings.

### 3.2. Characterization of the GABA-T Protein in Mulberry

To examine the potential role of the *GABA-T* gene in the mulberry response to salt stress, the gene was cloned from Husang 32 and designated as *MuGABA-T* (GenBank accession number OM289663), and a full length encoding cDNA of 1536 bp was obtained, which encodes a protein of 511 amino acids with a predicted molecular weight of 56.63 kDa and *p*I of 7.36. Multiple sequence alignments revealed that MuGABA-T protein has the conserved aminotransferase class-III domains of GABA-T proteins. Besides these conserved domains, the protein also contains eight conserved pyridoxal 5′-phosphate binding sites, which were essential for catalytic function and were involved in binding PLP at the active site (Figure 2). Phylogenetic analysis of GABA-T proteins from mulberry and other plants showed that the phylogenetic tree was obviously clustered into two branches of monocotyledons and dicotyledons. In the branches of dicotyledons, MuGABA-T and the proteins from other Rosaceae plants were clustered into one branch, indicating that they have the closest genetic relationship (Figure 3A). The 3D structure of MuGABA-T protein was predicted using SWISS–MODEL, and the results showed that it contained 57.75% random coil, 29.94% alpha helix and 8.22% extended strands (Figure 3B). In addition, the predictions of N-terminal extension and subcellular localization suggested that the protein contained no obvious signal peptide and was targeted to the chloroplast.

### 3.3. Expression Patterns of MuGABA-T

Firstly, the expression pattern of *MuGABA-T* gene in various mulberry tissues or organs was analyzed by qRT-PCR. The data showed that *MuGABA-T* was expressed ubiquitously in the investigated tissues, but it had higher expression levels in leaves and roots that in flowers and fruits (Figure 4A). In addition, the salt-induced expression pattern of *MuGABA-T* was explored and the results showed that its expression both in the roots and leaves was significantly induced by salt stress and reached a maximum level of expression 24 h after salt stress. Furthermore, the promoter of *MuGABA-T* gene was cloned and fused with *GUS* gene and was transiently expressed in tobacco leaves. The results of GUS staining showed that GUS activity was strongly induced by NaCl, ABA and SA but not obviously by MeJA (Figure 4B). Therefore, *MuGABA-T* may not only be involved in salt stress; its expression level may also be associated with ABA and SA.

### 3.4. Overexpression of MuGABA-T in Arabidopsis Enhances Salt Sensitivity

To explore the role of *MuGABA-T* in responses to salt stress, transgenic *MuGABA-T Arabidopsis* plants were generated. Firstly, comparisons of germination rates of transgenic *MuGABA-T* and wild-type (WT) plant seeds upon salt stress were performed. The results showed that the germination rates of *MuGABA-T*-overexpression plants showed no difference from those of the WT plants, and all the seeds germinated, reaching around 100% at the 7th day under the normal conditions. Under the treatment of 100 mmol·L^−1^ NaCl, the germination rate of WT and *MuGABA-T*-overexpression seeds was 84.7 and 64.7%, respectively. When the concentration of NaCl was increased to 150 mmol·L^−1^, only 34.0% of the *MuGABA-T*-overexpression seeds were able to be germinated, whereas 71.7% of the WT seeds germinated (Figure 5A,B). Similarly, the primary root growth of *MuGABA-T*-overexpression seedlings was also more sensitive to NaCl treatment than that of WT ones (Figure 5C). Under 100 mmol·L^−1^ NaCl treatment, the average root length of WT and *MuGABA-T*-overexpression seedlings was 0.80 and 0.52 cm, respectively. When the salt concentration was increased to 150 mmol·L^−1^, the average root length of WT seedlings was 0.34 cm, while the average root length of *MuGABA-T*-overexpression seedlings was only 0.16 cm (Figure 5D). These data presented above suggested that overexpression of *MuGABA-T* in *Arabidopsis* enhances salt sensitivity in the seed germination stage.

To further assess the potential role of *MuGABA-T* gene in response to salt stress, the WT and *MuGABA-T*-overexpressing seedlings were irrigated with 150 mmol·L^−1^ NaCl solution at the vegetable growth stage. Under normal growth conditions (controls), there was no significant difference in phenotype between the WT and transgenic *MuGABA-T* seedlings, while all the seedlings showed severe growth retardation under salt stress conditions. Most of the leaves of *MuGABA-T* transgenic seedlings turned yellow-green and showed necrosis; however, the leaves of WT seedlings remained green and showed fewer symptoms of damage (Figure 5E). In concordance with these results, the *MuGABA-T* transgenic plants showed significantly higher MDA content than the wild-type plants under salt stress (Figure 5F), indicating that the *MuGABA-T*-overexpression plants exhibited higher rates of cell damage and are more sensitive to salt stresses than WT plants. Overall, these data indicated that overexpression of *MuGABA-T* in *Arabidopsis* leads to increased sensitivity to NaCl.

### 3.5. Exogenous Application of GABA Alleviates the Sensitivity of Overexpressing MuGABA-T Arabidopsis to Salt Stress

Since GABA-T is the key component in GABA metabolism, to explore whether the sensitivity of overexpressing *MuGABA-T Arabidopsis* to salt stress is due to the decrease of GABA accumulation in the transgenic plants, analysis of GABA content in overexpressing *MuGABA-T* and WT seedlings was first performed, which showed that overexpressing *MuGABA-T* seedlings constitutively accumulated less GABA under normal growth conditions compared with WT seedlings (Figure 6A). Then, the transgenic *MuGABA-T* and WT seeds were grown on the solid MS medium supplemented with 150  mmol·L^−1^ NaCl and 0 or 50 mmol·L^−1^ GABA, respectively, and the germination rates were investigated. The results showed that GABA treatment had no significant effect on the germination rate of transgenic and WT seeds under the normal growth conditions. However, the application of exogenous GABA significantly improved the germination rate of transgenic and wild-type seeds under salt stress, especially the germination rate of transgenic seeds. Although the germination rates were still lower than those under normal conditions, there is no significant difference between wild-type and transgenic seeds (Figure 6B,C). Therefore, exogenous GABA can significantly alleviate the inhibitory effect of salt stress on the germination of transgenic *MuGABA-T* seeds.

In addition, the H_2_O_2_ and O_2_^−^ contents in WT and transgenic *MuGABA-T* seedlings under salt stress were detected with DAB and NBT methods. The results showed that the color depth of the transgenic *MuGABA-T* seedlings was significantly greater than that of the WT plants under salt stress without GABA; however, when these seedlings were treated with GABA, the accumulation of H_2_O_2_ and O_2_^−^ in transgenic *MuGABA-T* seedlings and wild-type seedlings decreased significantly, and there was no difference between WT and transgenic *MuGABA-T* seedlings (Figure 6D). Moreover, the SOD, POD and CAT activities were also detected, and the results showed that the activities of SOD, POD and CAT of the transgenic *MuGABA-T* and WT seedlings showed no significant differences under normal conditions. Under salt stress, these enzyme activities were evidently increased in the transgenic and WT seedlings, and they were higher in the WT seedlings than those in the transgenic ones. When the seedlings were sprayed with GABA, the activities of enzymes were evidently increased in all the seedlings, though their activities were higher in the WT seedlings than those in the transgenic ones (Figure 6E–G). Therefore, exogenous GABA application may be responsible for the increased activities of antioxidant enzymes and the enhanced reactive oxygen scavenging ability of the transgenic seedlings, and alleviate the sensitivity of overexpressing *MuGABA-T* seedlings to salt stress.

### 3.6. Overexpression of MuGABA-T in Mulberry Enhances Salt Sensitivity

Because the efficient genetic transformation and regeneration system of mulberry has not been established, to further explore the role of *MuGABA-T* in salt stress response, transgenic *MuGABA-T* mulberry plants carrying hairy roots were generated. GUS histochemical staining confirmed that the marker gene *GUS* was expressed in the hairy roots, and qRT-PCR analysis showed that the *MuGABA-T* gene was constitutively highly expressed in the hairy roots (Appendix A). After the confirmation of transgenic *MuGABA-T* hairy roots, the original roots were cut off and the plants carrying the transgenic hairy roots (PCTHR) were used for further analyses. Firstly, analysis of GABA content in the roots of PCTHR and WT was conducted and the result showed that the GABA content in the roots of PCTHR was much less than that in the roots of WT seedlings under normal growth conditions (Figure 7A). Then, these seedlings were subjected to salt stress to investigate their salt tolerance. One week post-NaCl-treatment, it was found that the growths of PCTHR were much more delayed than those of WT plants, and the seedlings showed serious stress damage symptoms, especially serious yellowing and wilting, while the wild-type plants showed only slight salt damage symptoms (Figure 7B). In addition, it was found that compared with the wild-type plants, the PCTHR exhibited remarkably higher levels of MDA content under salt conditions (Figure 7C). Moreover, H_2_O_2_ and O_2_^−^ contents under normal or stress conditions in WT and PCTHR were detected, and the results indicated that the DAB and NBT staining of all the plant leaves showed no differences under normal growth conditions. However, under the presence of 4‰ NaCl, the color depth of the leaves of PCTHR was significantly greater than that of the WT plants, indicating that the concentration of H_2_O_2_ and O_2_^−^ in the PCTHR was greater than that in the WT plants, and the PCTHR exhibited more ROS accumulation than WT plants under salt stress (Figure 7D). These observations were completely consistent with those obtained in *A**. thaliana*. Therefore, the overexpression of *MuGABA-T* in *Arabidopsis* or in mulberry hairy roots reduced the content of GABA in the transgenic plants, and the *MuGABA-T* gene may be involved in the ROS pathway in the response to salt stress and was responsible for a decrease in salt tolerance mediated by GABA.

## 4. Discussion

### 4.1. Overexpression of GABA-T Gene Results in Salt-Sensitive Phenotype

It has been reported that GABA is implicated in plant responses to multiple stresses, and GABA accumulation and metabolism were observed to be activated by salt treatment in a number of plant species [7,38,39]. *GABA-T* is the key component in GABA metabolism and catalyzes GABA conversion into succinic semialdehyde. The *Arabidopsis*
*GABA-T* mutant (*gaba-t* or *pop2*) is incapable of producing the GABA-T enzyme with biological activity [40], which will inhibit the degradation of GABA and lead to more GABA accumulation. Our results showed that overexpression of *MuGABA-T* in *Arabidopsis* or the hairy roots of mulberry resulted in less GABA accumulation (Figure 6A and Figure 7A). Interestingly, we found that overexpression of *MuGABA-T* in *Arabidopsis* and the hairy roots of mulberry caused the transgenic plants to exhibit enhanced sensitivity to salt stress. The results are contrary to the findings reported by Renault et al. (2010), which showed that the *pop2-1* was more oversensitive to salt stress than WT plants [25]. However, our results are consistent with the findings from Su et al. (2019) [26], in which the mutant *pop2-5* turned out to be tolerant to salt stress [26]. In addition, it was found that the glutamate decarboxylase (GAD1 and GAD2) mutant lines *gad1/gad2* were more sensitive to drought, but the triple mutant *gad1/gad2 × pop2-5* rescued the drought-sensitive phenotype of *gad1/gad2* [24]. Furthermore, the *pop2−1* mutant was also reported to be resistant to E-2-hexenal and Al stress [10,22]. All these results suggested that the degradation of GABA was inhibited in the pop mutants, and this will lead to more GABA accumulation in the plants and cause the mutants to have a more resistant phenotype. This is also consistent with our results that overexpression of *MuGABA-T* in *Arabidopsis* or the hairy roots of mulberry resulted in less GABA accumulation and caused the transgenic plants to exhibit enhanced sensitivity to salt stress. The reason for the different results from Renault et al. (2010) is not clear, and one reason might be that the mutant *pop2−1* was used by the authors while the overexpression of transgenic plants was used in our work. However, this does not seem to be the main reason for the difference, and the results by Renault et al. (2010) [25] may be an exception from the general trend that higher GABA levels are beneficial for salinity stress tolerance in plants.

### 4.2. GABA Accumulation Mitigates Salt Stress through ROS Pathway

Many studies have found that salt stress can accelerate production of ROS, resulting in oxidative stress and oxidative damage [41], and the ability to control ROS levels is highly correlated with plant stress tolerance [42,43]. It was proposed that GABA acts as a signal molecule that scavenges reactive oxygen species and regulates the activity of antioxidant enzymes, and there was evidence that GABA was associated with enhancing antioxidant metabolism and mitigating oxidative damage, which is a key regulatory pathway for improving stress tolerance of plants [44,45,46]. In *Caltha intermedia,* it was found that endogenous H_2_O_2_ was gradually increased in root and shoot tissue under salt stress, but exogenous application of GABA significantly reduced the level of H_2_O_2_ [47]. It was also shown that exogenous GABA treatment significantly increased the activities of POD and APX, and effectively reduced the production of H_2_O_2_ and alleviated the oxidative damage in leaves of *Lolium perenne* induced by hypoxia [48]. Moreover, it was reported that GABA can enhance antioxidant metabolism, reduce oxidative damage and improve drought tolerance of creeping bentgrass (*Agrostis stolonifera*) [49]. On the contrary, it was also found that the *gad1,2* mutant was unable to convert glutamate to GABA, and it accumulated more H_2_O_2_ under salt stress and displayed oversensitivity to salinity [26]. It has also been proven that the ROS level was significantly lower, while the antioxidant enzyme activities were higher in the white clover (*Trifolium repens* cv. Haifa) seeds pretreated with GABA than those without GABA pretreatment during germination under salt stress [50]. In our experiments, it was shown that there was less GABA accumulation in the PCTHR or transgenic *MuGABA-T Arabidopsis* plant than that in the WT plant. The transgenic *MuGABA-T* plants accumulated more H_2_O_2_ and O_2_^−^ than WT plants under salt stress, while the accumulation of H_2_O_2_ and O_2_^−^ were lower in the seedlings treated with GABA than those without GABA treatment. In addition, it was found that the activities of SOD, POD and CAT were evidently increased in all the seedlings applied with GABA. Therefore, it is possible that overexpression of *MuGABA-T* gene significantly reduced the accumulation of GABA, resulting in decreased ROS scavenging capability and oversensitivity to salt stress.

It was proposed that when carbon shortage becomes a limiting factor for plant growth and development, GABA can be used as a carbon source to support the TCA cycle under salt stress [23,51,52]. Since GABA-T catalyzes GABA conversion into succinic semialdehyde in the GABA shunt, its overexpression will accelerate the metabolism of GABA and provide more carbon sources for the TCA cycle, which may not have an adverse effect on the TCA cycle in the transgenic plants [26,53,54]. In addition, it was suggested that GABA is also involved in cytosolic pH and osmotic adjustment mechanisms defending against various stresses [55,56,57]. Moreover, it is suggested that GABA may act as a stress-induced intercellular signaling molecule involved in the signal transduction pathway associated with phytohormones [10,58]. Since the accumulation of GABA was decreased in the transgenic *MuGABA-T* plants (Figure 6A and Figure 7A), in this context, it cannot be excluded that GABA plays a role in *MuGABA-T*-mediated plant salt tolerance. Further studies are required to elucidate the precise mechanism of the effect of *GABA-T* on plant salt tolerance.

## 5. Conclusions

In conclusion, our data showed that MuGABA-T protein shares some conserved characteristics with its homologs, and its gene was constitutively expressed at different mulberry tissues and was substantially induced by NaCl, ABA and SA. Moreover, our results demonstrated that exogenous application of GABA significantly reduced the salt damage index and increased plant resistance to NaCl stress, and it was shown that the heterogeneous expression of *MuGABA-T* in *Arabidopsis* or overexpression of it in mulberry hairy roots reduced the content of GABA in the transgenic plants and enhanced the plants’ salt sensitivity. Particularly, *MuGABA-T* gene was found to play a pivotal function in mediating the rates of oxygen radical formation and detoxification and was responsible for a decrease in salt tolerance. Our findings help to further elucidate the function of the *GABA-T* gene and will facilitate genetic improvement in mulberry to improve its salt tolerance.

## Figures and Tables

**Figure 1 genes-13-00501-f001:**
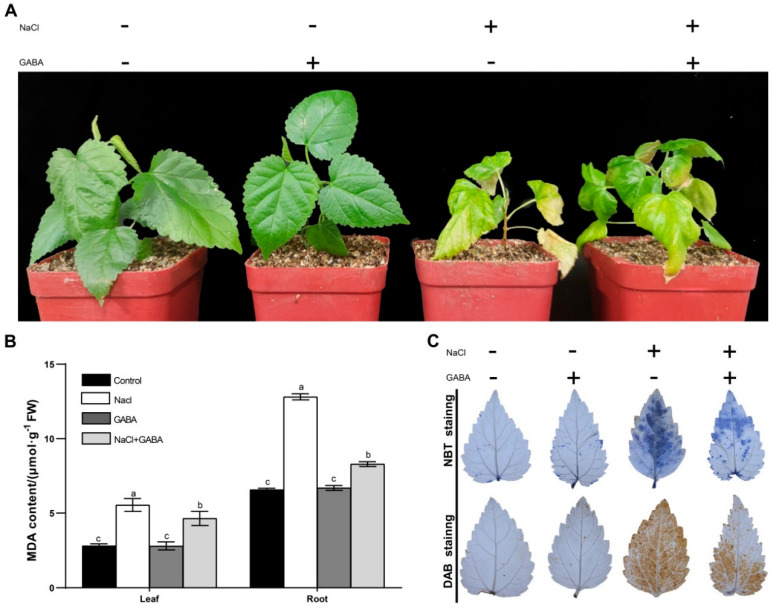
Exogenous application of gamma-aminobutyric acid (GABA) conferred increased tolerance of salt stress on mulberry seedlings. The stress treatments were completed on one-month-old seedlings. (**A**) Phenotype of one-month-old seedlings treated with 4‰ NaCl for 20 days with or without 50 mmol·L^−1^ GABA. (**B**) Malondialdehyde (MDA) contents in the leaves and roots of the seedlings under salt stresses. Values are the average of three biological replicates ±SD. Different letters above the columns indicate significant differences by Duncan’s multiple range test (*p* < 0.05). (**C**) Accumulation of superoxide and hydrogen peroxide (H_2_O_2_) in the leaves of mulberry seedlings. The purple-blue staining in the leaves indicates formazan deposits produced by superoxide anion reacting with nitroblue tetrazolium (NBT) and the brown coloration indicates the polymerization product formed by H_2_O_2_ reacting with 3,3′-diaminobenzidine (DAB). The leaf samples were collected at 24 h after stress treatment.

**Figure 2 genes-13-00501-f002:**
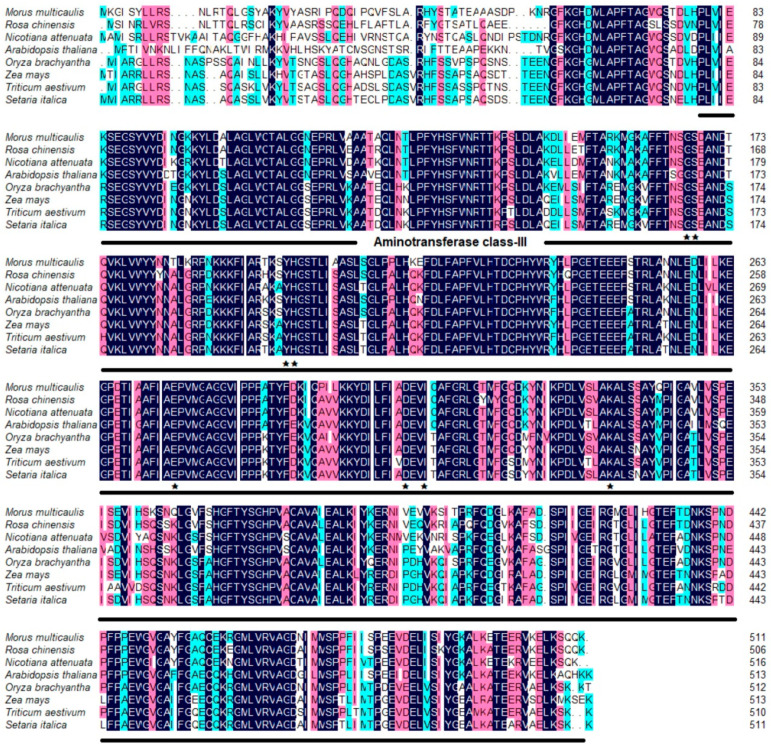
Alignment of the deduced amino acid sequences of MuGABA-T protein with GABA-transaminase (GABA-T) family members from other plants. The identical amino acid residues are shown in black and the similar amino acids are red shaded. Asterisks indicate the conservative pyridoxal 5′-phosphate binding sites. The heavy point indicates the conserved domain aminotransferase class-III. The aligned sequences include those proteins from *Morus multicaulis* (OM289663), *Rosa chinensis* (XP_024182390.1), *Nicotiana attenuate* (XP_019230459.1), *Arabidopsis thaliana* (NP_001189947.1), *Oryza brachyantha* (XP_006652817.1), *Zea mays* (ACF87978.1), *Triticum aestivum* (XP_044325747.1) and *Setaria italica* (XP_004976801.1).

**Figure 3 genes-13-00501-f003:**
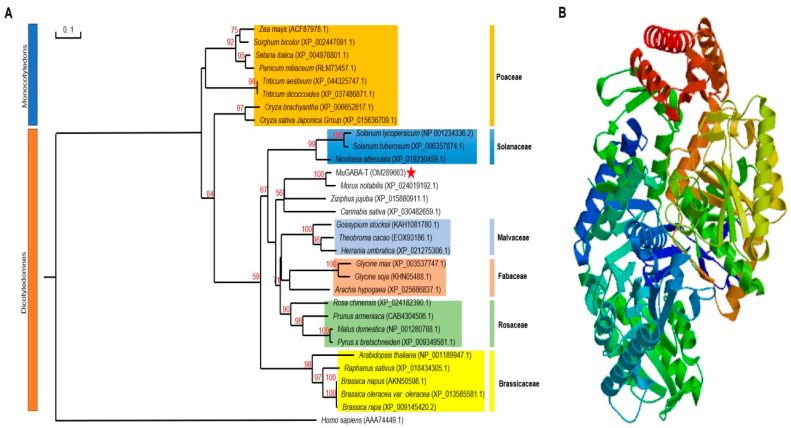
Phylogenetic analyses of GABA-T proteins from different plants and prediction of the three-dimensional structure of MuGABA-T protein. (**A**) The phylogenetic relationship of MuGABA-T and its homologous proteins. GenBank accession numbers of the proteins selected are shown in the brackets. Phylogenetic tree was generated using the neighbor-joining method with 1000 replicates. The bootstrap values are given on the nodes and the scale indicates genetic distance. (**B**) Three-dimensional structure of MuGABA-T proteins was built using SWISS-MODEL server by homology modeling.

**Figure 4 genes-13-00501-f004:**
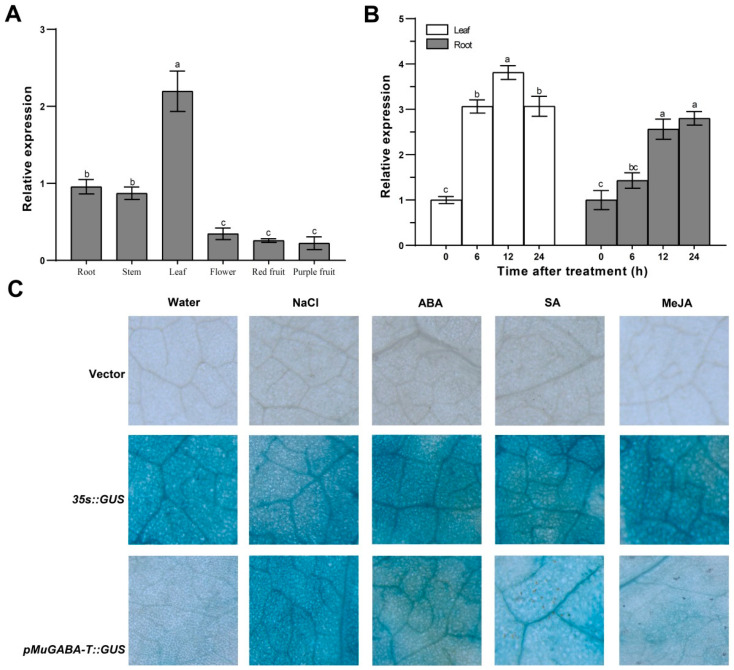
Expression pattern of *MuGABA-T* gene. (**A**) Expression pattern of *MuGABA-T* gene in different tissues. (**B**) Salt-stress-induced expression pattern of *MuGABA-T* gene in leaves and roots. The relative expression levels of *MuGABA-T* were calculated using the 2^−ΔΔCt^ method, and the Actin gene was used as a reference gene. Values represent the average of three biological replicates ±SD. Different letters above the columns indicate that the means in this column are significantly different (*p* < 0.05) according to Duncan’s multiple range test. (**C**) Transient expression pattern of *MuGABA-T* gene in *N. benthamiana* leaves. The leaves were infiltrated with the transformed *Agrobacterium* strains and sampled at 48 h after treatment with ABA, SA, MeJA or NaCl.

**Figure 5 genes-13-00501-f005:**
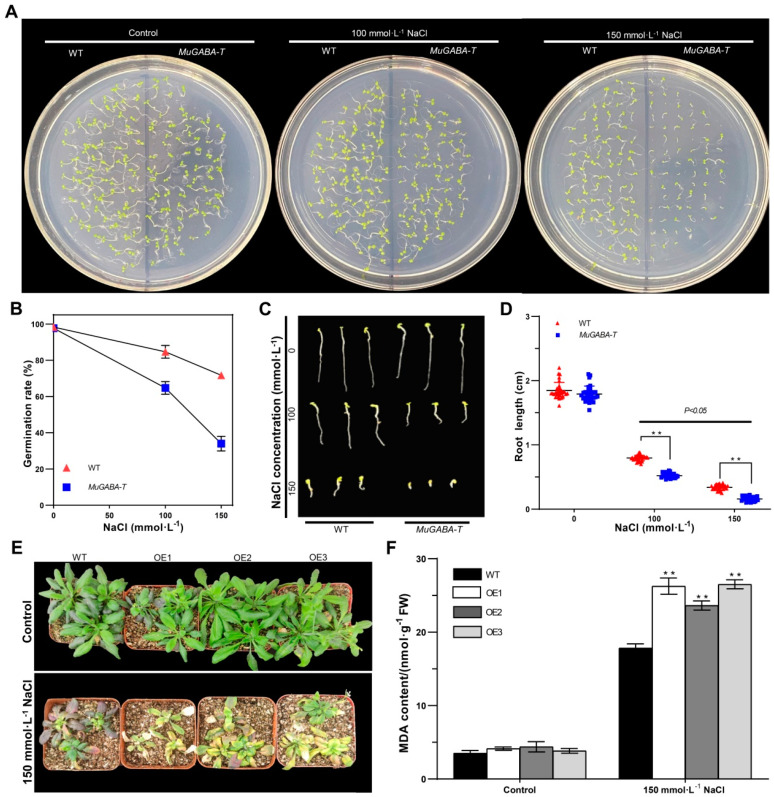
Phenotypic and physiological characterization of transgenic *MuGABAT Arabidopsis* upon NaCl treatment. Effects of NaCl stress on the germination (**A**,**B**), and root length (**C**,**D**) of WT and transgenic *MuGABA-T Arabidopsis*. Seeds were grown in Petri dishes for 7 d in 1/2 MS medium with 100 or 150 mmol·L^−1^ NaCl. Data are mean ± SD (n = 6), and the double asterisks indicate significant differences at *p* < 0.05. Effects of salt stress on morphology (**E**) and MDA content (**F**) of *Arabidopsis* seedlings. All the 4-week-old seedlings were grown in pots under control conditions, followed by salt stress treatment. Data are mean ± SD (n = 6), and the double asterisks indicate significant differences at *p* < 0.05.

**Figure 6 genes-13-00501-f006:**
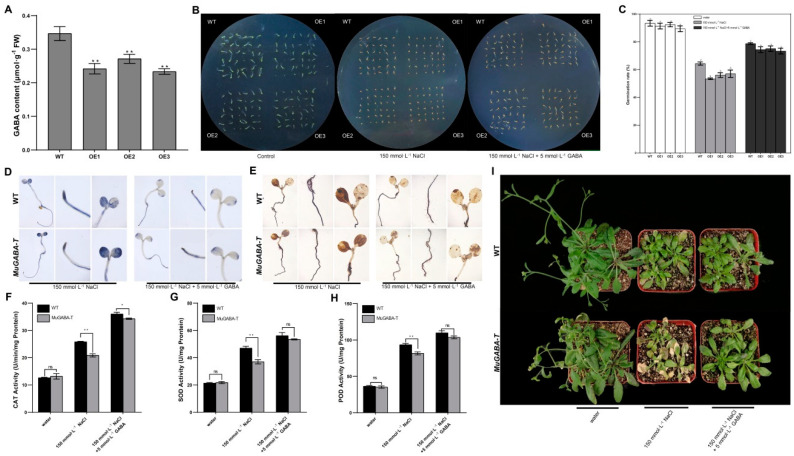
Exogenous application of GABA alleviates the sensitivity of transgenic *MuGABA-T Arabidopsis* to salt stress. (**A**) GABA content in transgenic *MuGABA-T* and WT *Arabidopsis* seedlings. The double asterisks indicate significant differences at *p* < 0.05. (**B**,**C**) Germination of WT and transgenic *MuGABA-T Arabidopsis* upon NaCl treatment with or without GABA. Seeds were grown in Petri dishes for 7 d in 1/2 MS medium upon 0 or 150 mmol·L^−1^ NaCl treatment with or without 5 mmol·L^−1^ GABA. Data are mean ± SD, and the different letters above the columns indicate significant difference (*p* < 0.05) according to Duncan’s multiple range test. Accumulation of superoxide (**D**) and H_2_O_2_ (**E**) in the *Arabidopsis* seedlings upon NaCl treatment with or without GABA. The purple-blue staining indicates formazan deposits produced by superoxide anion reacting with NBT and the brown coloration indicates the polymerization product formed by H_2_O_2_ reacting with DAB. Activities of superoxide dismutase (SOD) (**F**), peroxidase (POD) (**G**) and catalase (CAT) (**H**) in the seedlings upon NaCl treatment with or without GABA. The seedlings were collected at 7 d after treatment. Values are the means ± SD, and the double asterisks indicate significant differences at *p* < 0.05. (**I**) Morphology of *Arabidopsis* seedlings upon NaCl treatment with or without GABA. The 4-week-old seedlings were grown in pots under control conditions, followed by 150 mmol·L^−1^ NaCl treatment. At the same time, 50 mmol·L^−1^ GABA solution was sprayed on the leaves for 7 consecutive days, and the seedlings sprayed with water were used as controls.

**Figure 7 genes-13-00501-f007:**
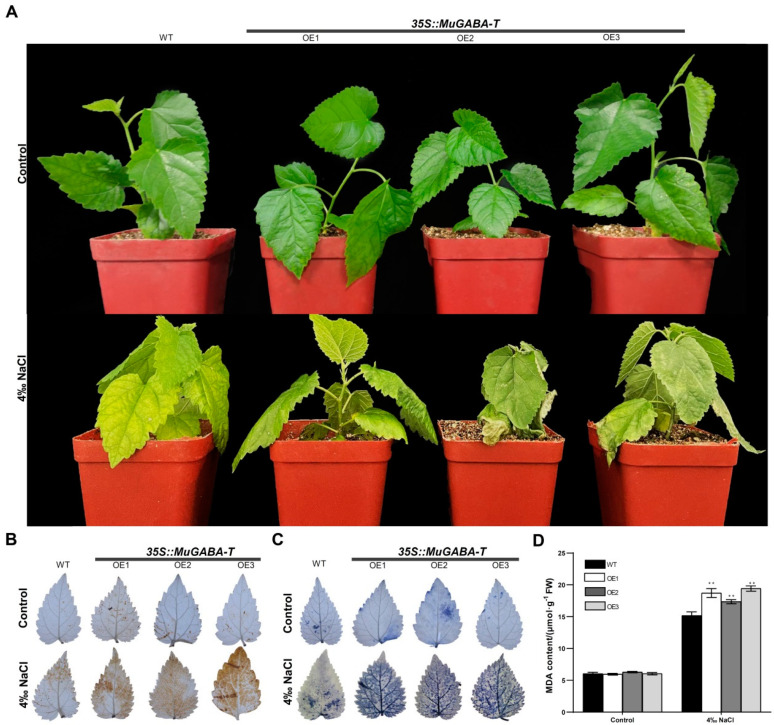
Salt sensitivity of the mulberry seedlings carrying the transgenic *MuGABAT* hairy roots. (**A**) Phenotypes of mulberry seedlings carrying hairy roots upon 4‰ NaCl treatment. Accumulation of superoxide (**B**) and H_2_O_2_ (**C**) in the mulberry seedlings carrying hairy roots upon NaCl treatment. The purple-blue staining indicates formazan deposits produced by superoxide anion reacting with NBT and the brown coloration indicates the polymerization product formed by H_2_O_2_ reacting with DAB. (**D**) MDA content in the leaves of the mulberry seedlings carrying hairy roots. Data are expressed as the average of three independent replicates ±SD, and the double asterisks indicate significant differences at *p* < 0.05. WT represents the mulberry seedlings carrying the transgenic empty vector hairy roots, and *35S::MuGABA-T* represents the mulberry seedlings carrying the transgenic *MuGABA-T* hairy roots. OE1-OE3: *MuGABA-T* transgenic lines.

## Data Availability

The datasets supporting the conclusions of this article are included within the article.

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
