# Peer review of "Characterization of GABA-Transaminase Gene from Mulberry (Morus multicaulis) and Its Role in Salt Stress Tolerance"

_genes, 2022, doi:10.3390/genes13030501_

Round 1

Reviewer 1 Report

Firstly, I take to thank the authors for this interesting paper with significant content and good presentation. I suggest to imporove the conclusions and to present more messages in this part of document .

Cordially 

Author Response

Q1. Firstly, I take to thank the authors for this interesting paper with significant content and good presentation. I suggest to improve the conclusions and to present more messages in this part of document.

Thank you for your comments. Based on your suggestion, some messages were added in the conclusions part, and we feel the revised manuscript is greatly improved as a result.

Reviewer 2 Report

The manuscript presented by Zhang et al. “Characterization of GABA-transaminase gene from mulberry (Morus multicaulis) and its role in salt stress tolerance” is significant for the scientific community. The authors worked on the GABA-T gene of mulberry, its molecular characterization expression profiles, and its functions in regulating defense responses against salt stresses.

The article was written well however, I found some faults, mistakes, or the leaving out of crucial details in the manuscript, which is discussed below.

  • Write the gene names or nay scientific name in italics, check it carefully.
  • Write the RNA extraction method in details.
  • How to select the particular gene for cloning?
  • Could you modify the methods for enzymes activity? If yes, then write in detail.
  • There are some other mistakes such as spacing in-between the words.
  • Add some recent references related to study
  • I have suggested the authors please read the manuscript carefully and do the needful things.

Author Response

Q1. Write the gene names or nay scientific name in italics, check it carefully.

Based on your suggestion, we have checked the manuscript carefully, and all the gene names or nay scientific name were written in italics in the revised manuscript.

Q2. Write the RNA extraction method in details.

Based on your suggestion, the RNA extraction method was described in details in the revised manuscript.

Q3. How to select the particular gene for cloning?

The specific PCR primers were designed based on our available mulberry transcriptome data for PCR amplification of the particular gene, and we have added this information in the revised manuscript.

Q4. Could you modify the methods for enzymes activity? If yes, then write in detail.

As it is difficult to obtain the experiment materials in a short time, we are sorry that we have not changed the determination method of enzyme activity.

Q5. There are some other mistakes such as spacing in-between the words.

We have checked the manuscript carefully and corrected all the errors we could find.

Q6. Add some recent references related to study

Based on your suggestion, some recent references related to study were added in the revised manuscript.